# Exploring seasonal and regional relationships between the Evaporative Stress Index and surface weather and soil moisture anomalies across the United States

Jason A. Otkin[1], Yafang Zhong[1], David Lorenz[2], Martha C. Anderson[3], and Christopher Hain[4]

[1]Cooperative Institute for Meteorological Satellite Studies, University of Wisconsin-Madison, USA
[2]Center for Climatic Research, University of Wisconsin-Madison, USA
[3]USDA Agricultural Research Service, Hydrology and Remote Sensing Laboratory, Beltsville, MD, USA
[4]Marshall Space Flight Center, NASA, Earth Science Branch, Huntsville, AL, USA

**Correspondence:** Jason A. Otkin (jasono@ssec.wisc.edu)

**Abstract.** This study uses correlation analyses to explore relationships between the satellite-derived Evaporative Stress Index (ESI) – which depicts standardized anomalies in an actual to reference evapotranspiration fraction – and various land and atmospheric variables that impact evapotranspiration. Correlations between the ESI and forcing variable anomalies calculated over sub-seasonal time scales were computed at weekly and monthly intervals during the growing season. Overall, the results

5   revealed that the ESI is most strongly correlated to anomalies in soil moisture and 2-m dew point depression. Correlations between the ESI and precipitation were also large across most of the U.S.; however, they were typically smaller than those associated with soil moisture and vapor pressure deficit. In contrast, correlations were much weaker for air temperature, wind speed, and radiation across most of the U.S., with the exception of the south-central U.S. where correlations were large for all variables at some point during the growing season. Together, these results indicate that changes in soil moisture and near-

10  surface atmospheric vapor pressure deficit are better predictors of the ESI than precipitation and air temperature anomalies are by themselves. Large regional and seasonal dependencies were also observed for each forcing variable. Each of the regional and seasonal correlation patterns were similar for ESI anomalies computed over 2-, 4-, and 8-week time periods; however, the maximum correlations increased as the ESI anomalies were computed over longer time periods and also shifted toward longer averaging periods for the forcing variables.

## 1  Introduction

High-resolution monitoring of vegetation health conditions using remote sensing observations provides valuable information that is widely used for a variety of purposes, such as drought monitoring (AghaKouchak et al., 2015), ecological health assessments (Li et al., 2014), and crop yield forecasting (Huang and Han, 2014; Johnson, 2016). Vegetation health and growth dynamics are influenced by a myriad of factors such as the timing and amount of rainfall, changes in evaporative demand due

20  to anomalous weather conditions, and the availability of sufficient root zone soil moisture to meet the vegetation's water requirements during different stages of its growth. In addition, the potential exposure to multiple or prolonged climate extremes

(for example, more than one year of drought) and other environmental factors such as insect infestations, disease, fires, and severe storms, influence the health and resiliency of vegetation.

In recent decades, numerous methods have been developed to extract information about various biophysical and physiological characteristics of vegetation using remote sensing observations from geostationary and low earth orbiting satellite sensors. Early studies exploited differences in the observed surface reflectance between red (visible) and near infrared bands in vegetated areas to derive a dimensionless estimate of plant vigor and standing biomass referred to as the normalized difference vegetation index (NDVI) (Tucker, 1979). The NDVI has been extensively used for global drought monitoring given its ability to identify regions containing poor vegetation health and its routine availability over many decades. Subsequent studies have developed newer vegetation indices derived from the NDVI such as the vegetation condition index (Liu and Kogan, 1996), enhanced vegetation index (Huete et al., 2002), and drought severity index (Mu et al., 2011). The availability of high quality surface reflectance observations from the Moderate-resolution Infrared Spectroradiometer (MODIS) sensor starting in 2000 led to the development of numerous products depicting various vegetation characteristics over the entire globe. These include quantities such as the fraction of photosynthetically active radiation, leaf area index, and gross primary productivity (Myneni et al., 2002; Heinsch et al., 2003). Other studies have used data from satellite sensors such as the Greenhouse gases Observing Satellite (GOSAT) to estimate solar-induced chlorophyll fluorescence (SIF) in terrestrial vegetation (Guanter et al., 2007; Frankenberg et al., 2011; Joiner et al., 2011; Sun et al., 2015). Though existing SIF datasets have coarse horizontal resolution, SIF is a useful metric because it provides direct information about the biochemical, physiological, and metabolic functioning of the plant canopy. Indeed, recent studies by Frankenberg et al. (2011) and Guanter et al. (2014) have shown that SIF is highly correlated to the gross primary production of terrestrial vegetation. In general, regions characterized by higher than normal values for any of these vegetation metrics will typically contain healthier and more productive vegetation.

Another measure that is widely used to monitor the overall health of vegetation is evapotranspiration (ET), which is a key component of terrestrial ecosystems because it links the carbon, energy, and hydrological cycles. Though ET represents the combination of evaporation from the surface and transpiration from vegetation, prior work has shown that transpiration is the dominant source of ET in vegetated areas (Budyko, 1974), thereby making it useful for monitoring moisture stress in vegetation. The rate of transpiration is controlled by the physiological characteristics of the plants, the amount of soil moisture in the root zone accessible by the plants, and various atmospheric factors, including the net radiation at the land surface, vapor pressure deficit over the leaf surface, and wind speed immediately above the land surface. ET rates are also influenced by the timing and amount of precipitation (or irrigation in managed landscapes) through its recharge of soil moisture. If sufficient root zone soil moisture is available, vegetation will typically increase its transpiration rate in response to elevated evaporative demand; however, once soil moisture content reaches the wilting point, vegetation will curtail its water usage. This transition from an energy-limited regime to a moisture-limited regime will result in an abrupt decrease in ET and the onset of moisture-related stress. Because ET anomalies are driven by multiple factors in addition to soil moisture status, it is often useful to compare the observed ET to a reference ET that can account for changes in the evaporative demand and solar radiation load. Then, if the ratio of the actual to reference ET – known as the reference ET fraction – is smaller (larger) than normal for a given location and time of year, this suggests that moisture-related stress in vegetation is higher (lower) than normal (Anderson et al., 2007a).

Various options are available for monitoring ET during the growing season. For example, direct measurements of ET can be obtained using flux tower networks such as AmeriFlux and FLUXNET (Baldocchi et al., 2001); however, their utility for large-scale monitoring is limited by their poor spatial sampling. High-resolution ET datasets can be generated using sophisticated land surface models such as those included in the North American Land Data Assimilation System (NLDAS) (Xia et al., 2012a, b). Though these datasets are spatially and temporally continuous, their accuracy will depend on the accuracy of the land surface models and the precipitation, atmospheric, soil property, and vegetation datasets that drive them (Beljaars et al., 1996). High-resolution ET datasets can also be generated using satellite observations by linking instantaneous ET rates to observables such as vegetation cover fraction and land surface temperature. Because ET estimates derived from infrared and visible satellite observations can only be computed when clouds do not obscure the surface, more complete domain coverage can be obtained by compositing clear-sky ET estimates over longer time periods (Anderson et al., 2013). Satellite-derived ET datasets covering regional and global domains can be obtained from a variety of sources, such as the MODIS Global Evapotranspiration Project (Mu et al., 2011), the Global Land Evaporation Amsterdam Model (Martens et al., 2017), and the Atmosphere Land Exchange Inverse (ALEXI) model used to compute the Evaporative Stress Index (ESI) (Anderson et al., 2007a, b, 2011).

Prior studies have shown that the ESI, representing standardized anomalies in the reference ET fraction, can provide early warning of drought development because vegetation often curtails its water usage before visible signs of moisture stress become evident in the vegetation (Otkin et al., 2013; Anderson et al., 2013). Though the ESI has primarily been used to monitor agricultural and ecological drought conditions (Anderson et al., 2007b, 2011; Otkin et al., 2013), and is well suited for the early detection of rapid onset flash drought events (Anderson et al., 2013; Otkin et al., 2015a, 2016, 2018), it can also be used to identify regions with healthy vegetation as inferred by higher than average ET rates. As such, it provides useful information about vegetation health under both favorable and unfavorable growing conditions and has been shown to have high correlations to agricultural crop yields (Anderson et al., 2016a, b; Otkin et al., 2016). Furthermore, Otkin et al. (2014, 2015a) have shown that unusually rapid decreases in the ESI provide useful information about the likelihood of drought development over the next 1-2 months, presumably due to soil moisture memory and its impact on vegetation. More recently, Lorenz et al. (2017a, b, 2018) developed a hybrid-statistical method that combines information from the ESI with precipitation and soil moisture anomalies to predict changes in the U.S. Drought Monitor (Svoboda et al., 2002) over sub-seasonal time scales. Their method had some forecast skill and was shown to provide useful forecasts, especially during flash drought events characterized by rapid intensification.

Given the drought monitoring capabilities of the ESI and the desire within the agricultural and natural resources communities for sub-seasonal drought intensification forecasts during the growing season (Otkin et al., 2015b), it is prudent to explore adaptation of the statistical method developed by Lorenz et al. (2017a, b) so that it can be used to predict changes in the ESI rather than the U.S. Drought Monitor because the ESI is a more direct measure of vegetation health. Such efforts would align with the increasing interest within the forecasting community to produce sub-seasonal forecasts that can fill the gap between medium-range weather forecasts and seasonal forecasts (Vitart et al., 2017). As a first step in this process, this study uses correlation analyses to examine relationships between the ESI and various land surface and atmospheric variables that control ET on sub-seasonal time scales. The study explores regional and sub-seasonal changes in the strengths of the correlations

using a version of the ESI that covers the contiguous U.S. (CONUS) with 4-km horizontal grid spacing. This study augments prior analyses by Anderson et al. (2013) and McEvoy et al. (2016) that examined correlations between the ESI and various soil moisture, precipitation, evaporative demand, and vegetation datasets over seasonal time scales. It also builds upon a recent study by Hobbins (2016) that used a variability attribution technique to assess the contribution of individual atmospheric drivers

on the regional and seasonal variability in reference ET across the U.S. Information from the correlation analyses performed during this study will inform efforts to develop sub-seasonal forecasts depicting changes in the ESI or similar quantities. The paper is organized as follows. Section 2 contains descriptions of the atmospheric, soil moisture, and ET datasets used during this study. Results from correlation analyses are shown in Section 3, with conclusions and a discussion provided in Section 4.

## 2   Data and Methodology

### 2.1   Evaporative Stress Index

The ESI shows standardized anomalies in a reference ET fraction ($ET/ET_{ref}$), where ET is the actual ET flux and $ET_{ref}$ is a reference ET flux computed using a Penman-Monteith formulation (Allen et al., 1998). Using a reference ET helps minimize the impact of the seasonal cycle in net radiation at the land surface when assessing anomalies in ET. As discussed in Anderson et al. (2007a), comparison of the observed ET flux to a reference ET flux provides a more meaningful depiction of moisture-

related stress than ET alone because it places changes in actual ET in context with observed changes in the evaporative demand and solar radiation forcing. For example, lower than normal ET does not necessarily mean that the vegetation is experiencing moisture stress if the evaporative demand is also lower.

The actual ET flux is estimated using the ALEXI model (Anderson et al., 2007a, 2011). ALEXI computes the ground, latent, and sensible heat fluxes for bare soil and vegetated components of the land surface using land surface temperatures retrieved

from satellite thermal infrared imagery and the Norman et al. (1995) two-source energy balance model. The partitioning of the surface energy budget into its constituent components is achieved through use of vegetation cover fraction estimates derived from the MODIS leaf area index product (Myneni et al., 2002). For each satellite pixel, the total surface energy budget is computed using the observed increase in land surface temperatures from  1.5 h after local sunrise until 1.5 h before local noon. The atmospheric boundary layer growth model developed by McNaughton and Spriggs (1986) is used to provide closure for

the energy balance equations, with temperature profiles in the lower troposphere used by the model obtained from the Climate Forecast System Reanalysis (CFSR) (Saha et al., 2010). The ALEXI model is run daily on a 4-km resolution grid covering CONUS using land surface temperature estimates derived from the Geostationary Operational Environmental Satellite (GOES) imager. The reader is referred to Anderson et al. (2007a, 2013) for a complete description of the ALEXI model.

Because ET estimates derived from satellite thermal infrared imagery can only be computed for pixels that remain entirely

clear during the morning integration period, the resultant daily ET datasets often have extensive data gaps. This issue is partially remedied by compositing the clear-sky ET estimates and corresponding reference ET fluxes over multi-week time periods. Standardized ET fraction anomalies, expressed as pseudo z-scores normalized to a mean of 0 and a standard deviation of 1, are then computed at weekly intervals using data composited over 2, 4, and 8 week time periods. The mean and standard

deviations for each week and compositing period are computed separately for each grid point using data from 2001-2015. Positive (negative) ESI anomalies depict above (below) normal reference ET fractions that typically correspond to better (worse) than average vegetation health and higher (lower) than average soil moisture content.

## 2.2 North American Land Data Assimilation System

The ESI anomalies were compared to modeled soil moisture anomalies computed using data from three NLDAS-2 models (Xia et al., 2012a, b), including the Noah (Ek et al., 2003; Barlage et al., 2010; Wei et al., 2013), Mosaic (Koster and Suarez, 1996), and Variable Infiltration Capacity (Liang et al., 1996) models. Each of these land surface models uses discretized forms of the energy and water balance equations to simulate changes in soil moisture content in multiple layers. Though each model uses the same atmospheric and precipitation forcing datasets, the soil moisture response can differ between models because

they may use different approximations to treat key processes such as evaporation, drainage, canopy uptake, and vegetation rooting depth. Given these differences, the ensemble average soil moisture is used here to represent the spatial distribution of soil moisture conditions. Xia et al. (2014) has shown that the ensemble average is more accurate than the individual models at depicting drought conditions. Ensemble mean soil moisture analyses generated each day for the topsoil (0-10 cm) and total column (0-200 cm) layers were averaged over 2-, 4-, and 8-week time periods and then standardized anomalies were computed

at weekly intervals for each layer (hereafter referred to as TS and TC, respectively) using data from 1979-2015. These datasets are useful for this study because they provide spatially and temporally continuous soil moisture information across the entire CONUS.

## 2.3 Standardized Precipitation Index

Standardized Precipitation Index (SPI) (McKee et al., 1993) anomalies were also computed over 4- and 8-week time periods to

assess the relationship between the ESI and precipitation. The SPI is a normalized variable such that anomalies greater (less) than zero indicate that the observed precipitation for a given location was more (less) than the climatological mean for a given period of time. Gridded precipitation analyses for 1948-2015 were obtained from the Climate Prediction Center gauge-based analysis of daily precipitation reports from cooperative observers and National Weather Service stations (Higgins et al., 2000), with the $0.25°$ resolution daily precipitation analyses interpolated to the ESI grid using a nearest neighbor approach. The daily

datasets were summed to create 4- and 8-week accumulated precipitation amounts prior to computing the SPI.

## 2.4 Atmospheric variables

The relationships between the ESI and near-surface atmospheric conditions were evaluated using analyses from the CFSR, which is a fully coupled atmosphere-land-ocean modeling system (Saha et al., 2010). Given their importance for driving changes in ET, this study focuses on 2-m temperature, 2-m dew point depression, 10-m wind speed, and downwelling shortwave

radiation (hereafter referred to as TEMP, DPD, WSPD, and DSW, respectively). Daily averages were computed for each variable using analyses available every 6 h on a 38 km resolution grid, and then interpolated to the ESI grid using a nearest

neighbor approach. Standardized anomalies were computed at weekly intervals for 2-, 4-, and 8-week averaging periods using data from 1979-2015. It should be noted that using different baseline periods when computing the ESI, soil moisture, SPI, and atmospheric anomalies introduces some uncertainty in the correlation analyses presented in Section 3 because of potential trends in the data (especially for air temperature) during their respective periods of record. However, because this study focuses

on temporal correlations at the grid point scale, the results should not be strongly affected by the baseline period because the anomaly computation is simply a linear rescaling at each grid point. In addition, it is important to note that though all of ~~these~~ the atmospheric variables are derived from model output, they are constrained through the assimilation of satellite and conventional observations within the CFSR data assimilation system. Regional verification studies, such as those performed by Bao and Zhang (2013), Lindsay et al. (2014), Sharp et al. (2015) and Essou et al. (2016), have shown that the accuracy of the

CFSR near-surface variables are comparable to those from other reanalysis datasets and represent an important improvement over previous generations of reanalysis datasets. Fuka et al. (2013) have shown that when CFSR data was used to force a watershed model, that it produced stream discharge simulations that were as good or better than models forced using weather station observations. The use of reanalysis data introduces some uncertainty to the evaluation performed during this study but it has the advantage of providing uniform spatial resolution across the entire region.

**3  Results**

### 3.1  Monthly correlation analysis

In this section, the relationship between the ESI and various atmospheric and land surface variables is assessed during the warm season using correlation analyses. Figures 1 and 2 show the Pearson correlation coefficients between the 4-wk ESI and the corresponding 4-wk SPI, TS, TC, DPD, TEMP, WSPD, and DSW anomalies at monthly intervals from April to September.

Note that the sign is reversed for the DPD, TEMP, WSPD, and DSW correlations given the expectation that larger (smaller) values for each of these variables will typically be associated with higher (lower) moisture stress and negative (positive) ESI anomalies when assessed over long time periods. The correlations were computed separately for each grid point and month using all of the weekly analyses from 2001-2015 for which the end of the 4-wk period fell within a given month. This means that the sample size (n) for each grid point is equal to 60 or 75 depending upon whether a given month contains the end dates

for four or five of these 4-week periods. The statistical significance of the correlation at each grid point was determined using the "rtest" routine in the NCAR Command Language (NCL) package, with the probability value (p) set to 0.1. Using 4-wk periods for all of the datasets allows us to examine the contemporaneous relationships between the ESI and the various forcing variables across the entire U.S. during different portions of the growing season. The 4-wk ESI was chosen for this part of the analysis because it provides a balance between the fast response of the ESI to changing conditions when it is computed over

short time periods and the seasonal moisture stress signals contained in longer-term ESI anomalies (Otkin et al., 2013). A more comprehensive assessment of the relationships between the various forcing variables and ESI anomalies computed using different compositing lengths is presented in the regional analysis shown in Sec. 3.2.

Inspection of Figs. 1 and 2 reveals that in most locations the strongest correlations occur for the DPD, TS, TC, and SPI variables. This combination indicates that anomalies in the ESI are most closely related to anomalies in soil moisture and near surface humidity. The correlations for these variables show that periods characterized by larger (smaller) DPD and below (above) average TS, TC, and SPI often contain negative (positive) ESI values. In contrast, correlations for TEMP, WSPD, and DSW are much weaker across most of the U.S. with the exception of the south central U.S. where correlations are large for each of these variables at some point during the growing season. This region is located within an east-west transition zone between arid climates to the west and humid climates to the east where longitudinal shifts in the rainfall gradient strongly impact the weather. It is also a well-known hot spot for land-atmosphere coupling, which occurs when soil moisture and vegetation anomalies influence the partitioning of surface energy between sensible and latent heat fluxes (Koster et al., 2004). The results also show that the strengths of these relationships vary during the growing season across this region. For example, the correlations for DSW are largest during the spring and early summer when positive insolation anomalies due to reduced cloud cover can drive rapid evaporative loss, leading quickly to lower than normal ET fractions, whereas TEMP anomalies are more important during the second half of the growing season when unusually hot (cool) temperatures may hasten (delay) vegetation stress and senescence. For the remaining variables (DPD, TS, TC, and SPI), the correlations are large during most of the growing season. Together, this indicates that ET fraction anomalies within this region of enhanced land-atmosphere coupling are most closely related to variables capturing changes in the supply and demand of surface moisture.

Unlike the south central U.S. where statistically significant correlations exist between the ESI and each of the variables, much weaker correlations occur across other parts of the U.S. For example, very low ($< 0.2$) non-significant correlations predominate across most of the northeastern U.S. during the spring and early summer. The strength of the correlations increases during the second half of the growing season, with the largest correlations found for DPD, SPI, TS, and TC; however, they remain weaker than those found across the south central U.S. A similar evolution occurs within an east-west band extending from the Pacific Northwest to the Great Lakes, with the lowest correlations generally occurring in regions containing extensive forests. The low correlations indicate that there are no dominant drivers of normalized ET during the first half of the growing season in these regions, presumably because of their relatively cool and moist climates and the much deeper root structures in forests that allow trees to tap into deeper soil moisture than other types of vegetation. ET becomes more strongly coupled to the atmospheric and land surface variables later in the growing season as these regions move from being primarily energy-limited regimes to potentially moisture-limited regimes.

Finally, the correlations over most of California have a distinct seasonal cycle that is opposite that found in the heavily forested areas across the northern U.S. For example, with the exception of WSPD, all of the variables are strongly correlated ($>$ 0.6) to the ESI during the spring. These correlations then rapidly decrease after June and are generally non-significant across most of the state during August and September. This sequence suggests that ET is strongly influenced by the amount of cool season precipitation, which is a key component of the hydrological cycle in the region (Neiman et al., 2008), and its subsequent impact on soil moisture during the first half of the growing season. The results indicate that a wetter (drier) than normal cool season that then leads to wetter (drier) than normal soil moisture conditions during the spring are typically associated with

above (below) normal ET. The results also show that anomalies in evaporative demand strongly impact ET during this time period as evidenced by the large correlations between the ESI and the DPD, TEMP, and DSW variables from April to June.

## 3.2 Regional correlation analysis

This section assesses changes in the correlations between the ESI and the various forcing variables at weekly intervals from March to October when different compositing and averaging lengths are used to compute the anomalies for each variable. This analysis will be used to assess relationships between the ESI computed over short-to-intermediate time scales (2, 4, and 8 weeks) and each of the forcing variables computed over similar time periods. To capture regional differences in the relationships, the correlations were computed separately for the western, south central, north central, and eastern U.S. using data from 2001-2015. The outlines for each region are shown in Fig. 3. These regions were chosen based on geography, climate, and inspection of the spatial patterns in the correlations found in Figs. 1 and 2. In particular, the central U.S. regions encompass the meridional gradient between more arid climates to the west and more humid climates to the east (Seager et al., 2018). This area was further separated into south-central and north-central regions to highlight the stronger correlations over the south-central U.S. and to account for large regional differences in the variance of the atmospheric drivers of reference ET noted by Hobbins (2016). Likewise, remaining areas were simply grouped into western and eastern regions that generally encompass more arid and more humid climates, respectively. By assessing the relationships over such large regions, some of the local details discussed in Section 2.1 will be lost; however, this approach makes discussion of the results more tractable while still preserving the most dominant signals within these larger regions.

Figure 4 shows correlations between the 2-wk ESI anomalies and the various land surface and atmospheric variables for each region. The stippling on the figure denotes the weeks and variables for which at least half of the grid points in each region had a statistically significant correlation. Note, however, that the correlations were computed using all grid points in a given region, including those that had non-significant correlations. As was shown in the previous section, the strength of the relationship between the ESI and a given variable varies greatly across different parts of the U.S., with large sub-seasonal fluctuations in the correlations also apparent in many of the variables. For example, much lower and generally non-significant correlations occur in the TS, DPD, DSW, and SPI variables across the western U.S. from the middle of July to the end of August. Because this time period corresponds to the climatological peak of the North American Monsoon (Adams and Comrie, 1997), it is possible that the weaker correlations are at least partially due to the impacts of this climate feature on soil moisture and near-surface atmospheric conditions across the region. Other examples of large sub-seasonal fluctuations include the much larger TEMP correlations from June to September and the non-significant DSW correlations from mid-July to September across the south-central U.S. This pattern also occurs to a lesser extent across the north-central U.S. The larger impact of air temperature on ET during the summer across the central U.S. could be associated with the higher likelihood that daytime temperatures will exceed the heat tolerance of the vegetation. Likewise, in a predominantly moisture-limited regime during the second half of summer, changes in cloud cover and its impact on DSW become less important across this region.

Comparison of the correlations for a given variable reveals a complex relationship where the maximum correlations for some variables occur when anomalies are computed over short time periods (e.g., 2 weeks), whereas in others, the correlations

become larger as the averaging period increases. For example, the largest correlations for DSW, TEMP, and SPI occur for anomalies computed over an 8-wk time period. In contrast, the largest correlations for soil moisture occur when the TS and TC anomalies are computed over 2- and 4-wk time periods. Meanwhile, the correlations for DPD and WSPD are similar for all averaging periods. These patterns are evident in each region, but are most conspicuous across the south-central U.S., and remain consistent throughout the entire growing season. Together, these results indicate that ESI anomalies computed over short time periods are most strongly influenced by short-to-intermediate fluctuations in soil moisture that in turn are controlled by precipitation and evaporative demand anomalies occurring over longer time periods. The more pronounced transition from large to small correlations across the western and central U.S. as the averaging period increases for the TC and TS variables also shows that short-term ET anomalies in these drier areas are fundamentally tied to the availability of soil moisture over short time periods.

To assess how these relationships change when ESI anomalies are computed over longer time periods, Figs. 5 and 6 show correlations between each variable and the 4- and 8-wk ESI, respectively. Comparison to the 2-wk ESI correlations shown in Fig. 4 reveals that the seasonal patterns in the correlations remain similar for each region and variable when the ESI anomalies are computed over longer time periods; however, the maximum correlations shift toward longer averaging periods for most of the variables. This shift shows that the longer duration ESI anomalies are most closely related to atmospheric and land surface anomalies occurring over similarly long time scales, while still having some sensitivity to shorter fluctuations in these variables. The maximum correlation for a given variable also tends to increase as the ESI compositing period increases from 2 to 8 weeks, with the DPD, TS, and TC variables having the largest correlations in each region during most of the growing season regardless of the length of time used to compute the ESI.

An interesting pattern emerges when comparing the correlations for the DPD, TS, and TC variables. Whereas all three variables had their largest correlations to the 2-wk ESI when their anomalies were computed over 2- and 4-wk time periods, their behavior diverges for the 8-wk ESI for which the maximum correlations shift to longer averaging periods for the TS and DPD variables but remain large for TC regardless of the length of time used to compute anomalies for that variable. This pattern occurs in all of the regions, and shows that as the ESI anomalies are computed over longer time periods, they become most closely related to TS and DPD anomalies occurring over similar time scales, but to TC anomalies occurring over all time scales. This behavior is likely due to the tendency for TC soil moisture to change more slowly than DPD and TS soil moisture – both of which are more strongly influenced by synoptic-scale (e.g. weekly) weather features – and thus remain closely related to the ESI over multiple time scales.

As was the case with the 2-wk ESI, longer-term ESI anomalies are most strongly correlated to DSW across the central and western U.S. during the first half of the growing season. Likewise, SPI and WSPD correlations continue to be the largest over the south-central U.S, with weaker correlations found elsewhere. The relationship between the ESI and TEMP also remains strong over the south-central U.S., where correlations exceed 0.5 from June until the middle of September. In all of the regions, the TEMP correlations for a given averaging period are generally non-significant and smaller than the corresponding DPD correlations during the entire growing season for the 2-, 4-, and 8-wk ESI. This indicates that TEMP anomalies are not a dominant driver of changes in ET; rather, it is near surface humidity that is most important. For example, a period characterized

by hot temperatures may not necessarily lead to increased moisture stress (e.g., negative ESI) if it also accompanied by heavy rainfall, which is a common occurrence across the central and eastern U.S. during the summer. Instead, if hot temperatures occur alongside lower dew point temperatures, the resultant increase in the DPD will have a larger impact on ET than the higher TEMP alone. The stronger relationship between the ESI and DPD is consistent with prior work that has shown that stomatal conductance and the release of ET by many plant species is strongly controlled by the vapor pressure deficit (Oren et al., 1999). These results also suggest that drought forecasts that rely upon monthly-to-seasonal temperature outlooks to predict changes in vegetation health may be more accurate if anomalies in near surface humidity are also considered.

## 4  Conclusions and discussion

This study used correlation analyses to explore relationships between the satellite-derived ESI – which depicts anomalies in an actual to reference ET fraction – and a set of land and atmospheric variables that are known to influence ET through their impact on soil moisture and evaporative demand. Overall, the results showed that anomalies in ET as expressed by the ESI are most strongly correlated to anomalies in soil moisture and near-surface humidity (TS, TC, and DPD) regardless of the time period over which the anomalies are computed. Correlations between the ESI and precipitation (SPI) are also relatively large across most of the U.S.; however, they are typically smaller than the TS, TC, and DPD correlations for a given location and time of year. The strong correlations to soil moisture over sub-seasonal time scales are consistent with the seasonal correlation analyses described in Anderson et al. (2011, 2013). In contrast, correlations are relatively weak and often non-significant for TEMP, WSPD, and DSW across most of the U.S., except for the south-central U.S. where correlations are strong for all of the variables at some point during the growing season. The larger correlations in this region of enhanced land-atmosphere coupling are consistent with prior studies that have shown that the strength of the coupling is influenced by a wide variety of atmospheric and land surface processes. Unlike the south-central U.S., the correlations are much weaker in forested regions across the northern tier of the U.S. The correlations are especially weak during the spring and then increase during the second half of the growing season. The weak relationships to the ESI indicate that there are no dominant drivers of ET during the first half of the growing season in these northern locations; however, ET becomes more strongly coupled to the forcing variables later in the growing season as the regions transition from energy-limited regimes to potentially moisture-limited regimes.

Large sub-seasonal fluctuations in the correlations are also evident in some of the variables across parts of the U.S. For example, correlations to DSW are large across the central U.S. during the spring and early summer, whereas TEMP anomalies become more important during the second half of the growing season. Over the western U.S., the correlations are much lower for all variables during the climatological peak of the North American Monsoon during July and August when compared to other parts of the year. California also has a distinct seasonal pattern where the correlations are largest during the spring and then rapidly diminish after June. Each of the regional and seasonal patterns were similar for ESI anomalies computed over 2-, 4-, and 8-wk time periods; however, the maximum correlations typically increased as the ESI anomalies were computed over longer time periods and also shifted toward longer averaging periods for the forcing variables. This shift shows that ESI anomalies computed over short (long) time periods are most strongly correlated to atmospheric and land surface anomalies

occurring over similar time scales, while also having some sensitivity to anomalous conditions occurring over longer (shorter) time periods.

Overall, the large regional and seasonal variability in the correlation patterns found during this study are similar to the analysis presented by McEvoy et al. (2016) that assessed the relationship between the ESI and evaporative demand as expressed by the Evaporative Demand Drought Index (Hobbins et al., 2016). Their study showed that the largest correlations occurred over the south-central U.S., with the smallest correlations generally occurring across the northern U.S. during the spring. The broadly similar results in both studies demonstrate the important role that atmospheric evaporative demand has in driving changes in actual ET; however, the strong correlations between the ESI and soil moisture found in this study also illustrate that the vegetation response to anomalies in evaporative demand are strongly influenced by soil moisture. For example, even if the atmospheric demand is higher than normal, vegetation stress may not occur if sufficient soil moisture is available to meet the increased demand. These results show that it is important to monitor anomalies not only in atmospheric demand but also in soil moisture when assessing actual stress in vegetation.

Investigation of the monthly and regional correlations also showed that anomalies in the ESI are typically much more strongly correlated to anomalies in the DPD than they are to anomalies in TEMP during the entire growing season across most of the U.S. This indicates that in most situations it is the vapor pressure deficit rather than air temperature that is the most important driver of changes in ET ; however, it is reasonable to expect that large (small) vapor pressure deficit anomalies are more likely to occur when air temperatures are hotter (cooler) than normal. Likewise, though the correlations for SPI are relatively large, they are still generally smaller than those associated with the soil moisture variables. Together, these results indicate that fluctuations in soil moisture and near-surface humidity are better predictors of the ESI than are SPI and TEMP anomalies by themselves. This is consistent with a recent study by Ford and Labosier (2018) that showed that temperature and rainfall departures by themselves were only weakly related to the occurrence of rapid-onset flash droughts, whereas variables that explicitly account for changes in soil moisture content and near-surface humidity were more closely linked to the development of these features. It is also consistent with studies by Irmak et al. (2006) and Hobbins (2016) that showed that surface temperature is not always the most important driver of temporal variability in reference ET. Their studies revealed that other variables such as net radiation, wind speed, and water vapor mixing ratio, can have a larger impact than temperature on reference ET and that the relative importance of each variable changes during the year and across different regions.

Together, these findings also illustrate that existing monthly-to-seasonal outlooks that tend to focus on predicting anomalies in air temperature and precipitation are insufficient for predicting changes in agricultural or ecological drought conditions. Instead, greater focus should be placed on predicting changes in soil moisture and vapor pressure deficit given their more dominant influence on ET. Indeed, a recent study by Lorenz et al. (2018) has shown that inclusion of vapor pressure deficit and soil moisture predictions from climate models increased the accuracy of sub-seasonal drought intensification forecasts generated using a hybrid statistical method. This is also supported by a study by Seager et al. (2015) that showed that large forest fires are often associated with very large vapor pressure deficits caused by antecedent surface drying and large-scale subsidence.

*Competing interests.* The authors declare that there are no competing interests with the work performed during this study.

*Acknowledgements.* The authors would like to acknowledge support provided by the NOAA Climate Program Office (CPO) Modeling, Analysis, Predictions, and Projections program under grant NA14OAR4310226 and the NOAA CPO Sectoral Applications Research Program under grant NA16OAR4310130.

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

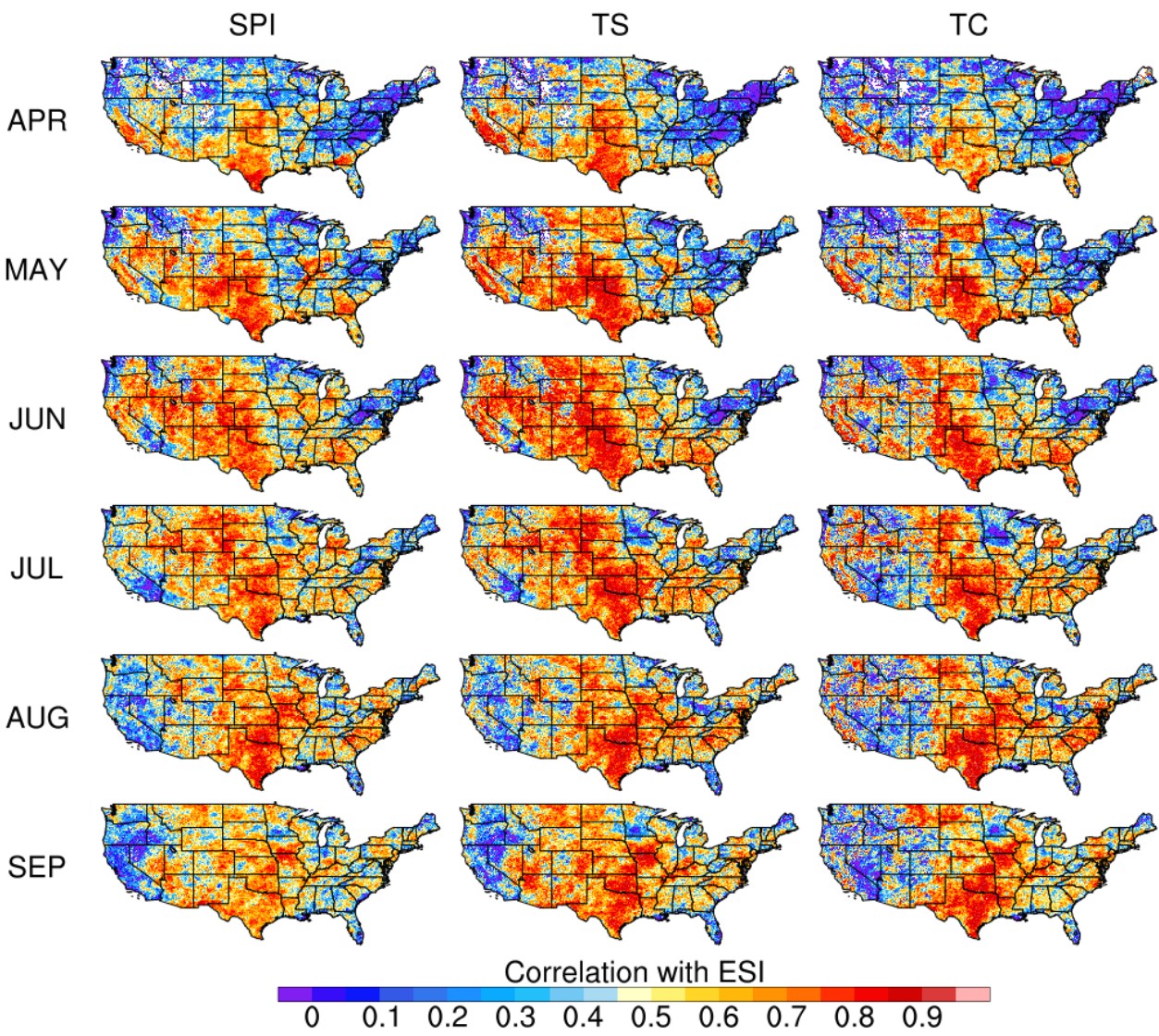

**Figure 1.** Pearson correlation coefficients between ESI anomalies computed over a 4-wk time period and SPI, TS, and TC anomalies computed over a 4-wk time period. The correlations were computed separately for each grid point and month using all of the weekly analyses from 2001-2015 for which the end of the 4-wk period was within a given month. Correlations > 0.21 are significant at the p=0.1 level.

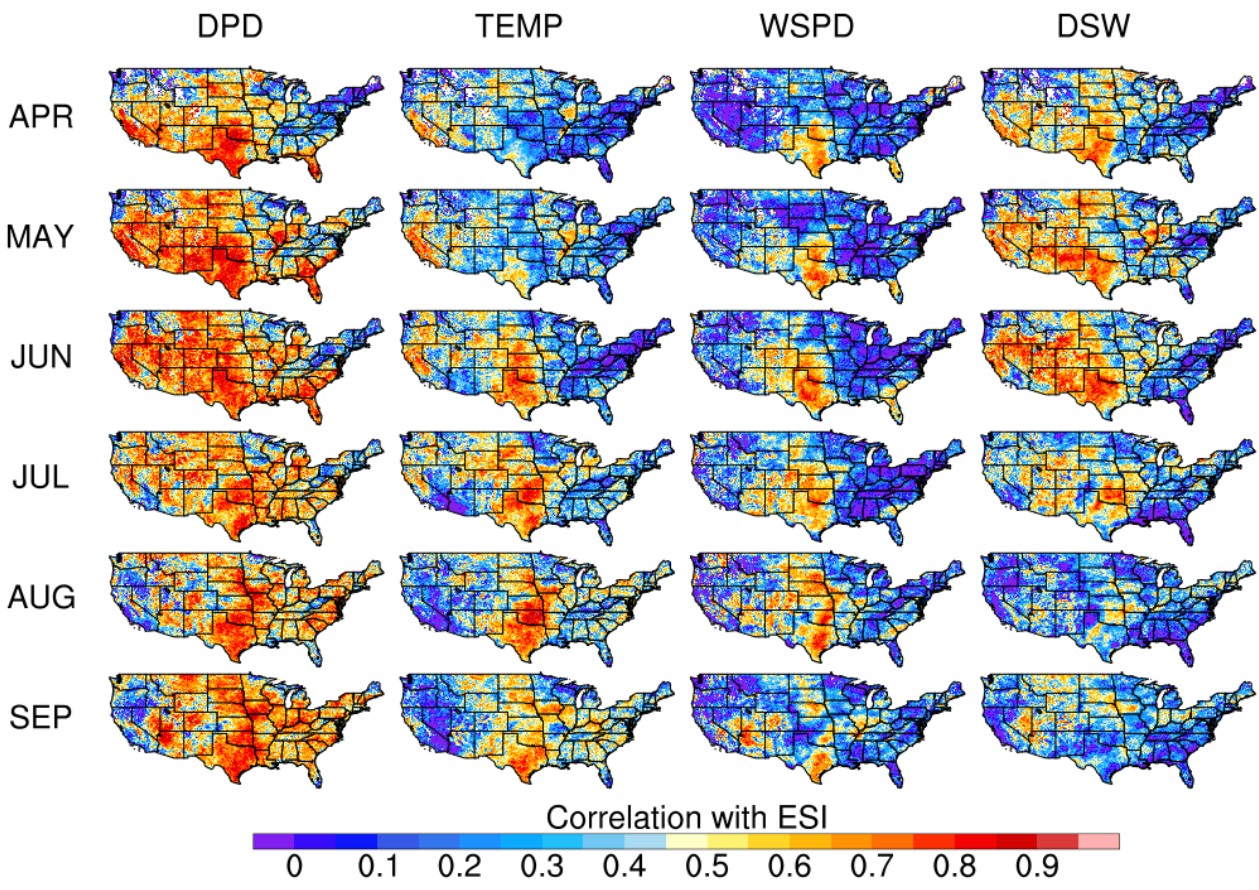

**Figure 2.** Same as Fig. 1, except for showing correlations between the ESI anomalies computed over a 4-wk time period and DPD, TEMP, WSPD, and DSW anomalies computed over a 4-wk time period. Note that the sign has been reversed for the DPD, TEMP, WSPD, and DSW correlations so that positive correlations indicative of enhanced drying are shown in yellow and red colors. Correlations > 0.21 are significant at the p=0.1 level.

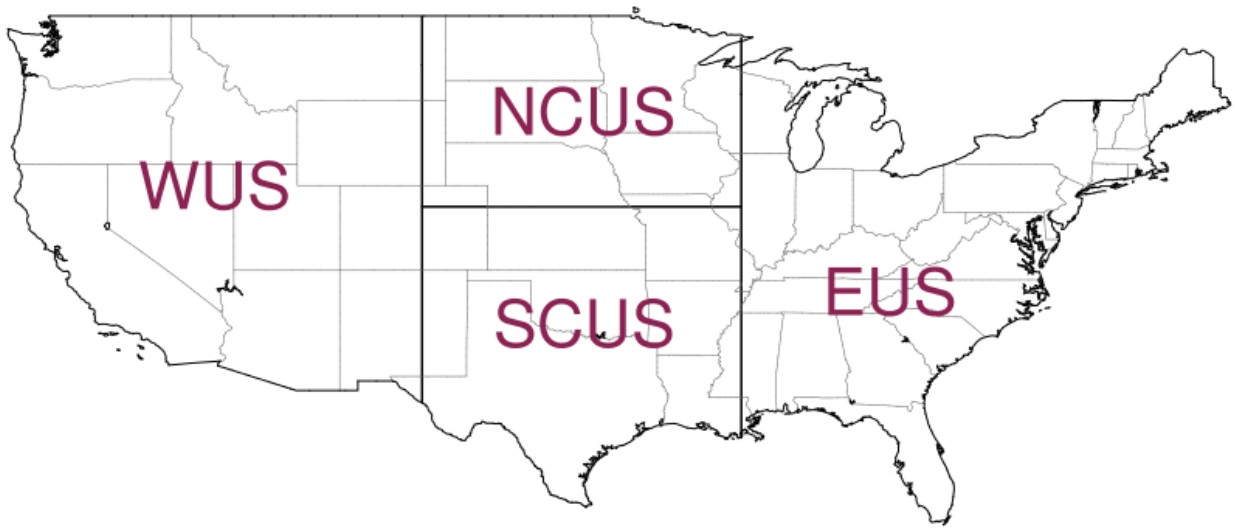

**Figure 3.** Geographic domain covered by each of the regions included in Figs. 4-6.

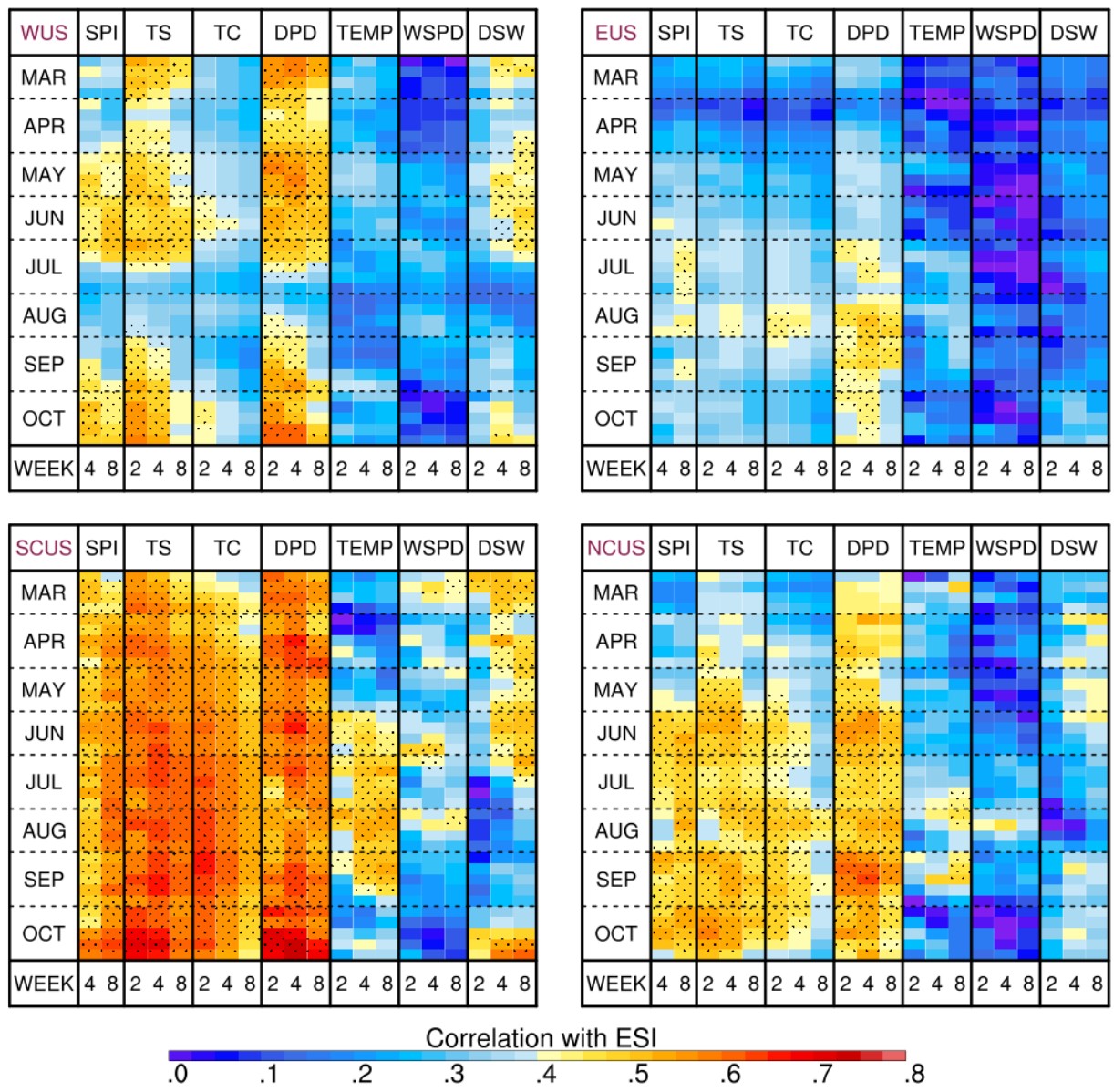

**Figure 4.** Weekly time series of correlations between the 2-wk ESI and each forcing variable computed using all land grid points in the western (WUS), eastern (EUS), south central (SCUS), and north central (NCUS) U.S. and data from 2001-2015. Correlations to the 4-, 8-, and 12-wk SPI are shown in columns 1-3, with correlations to the 2-, 4-, and 8-wk TS, TC, DPD, TEMP, WSPD, and DSW variables shown in columns 4-6, 7-9, 10-12, 13-15, 16-18, and 19-21, respectively. Note that the sign has been reversed for the DPD, TEMP, WSPD, and DSW correlations so that positive correlations indicative of enhanced drying are shown in yellow and red colors. The stippling denotes the weeks and variables for which at least half of the grid points in a given region had a statistically significant correlation at the p=0.1 level.

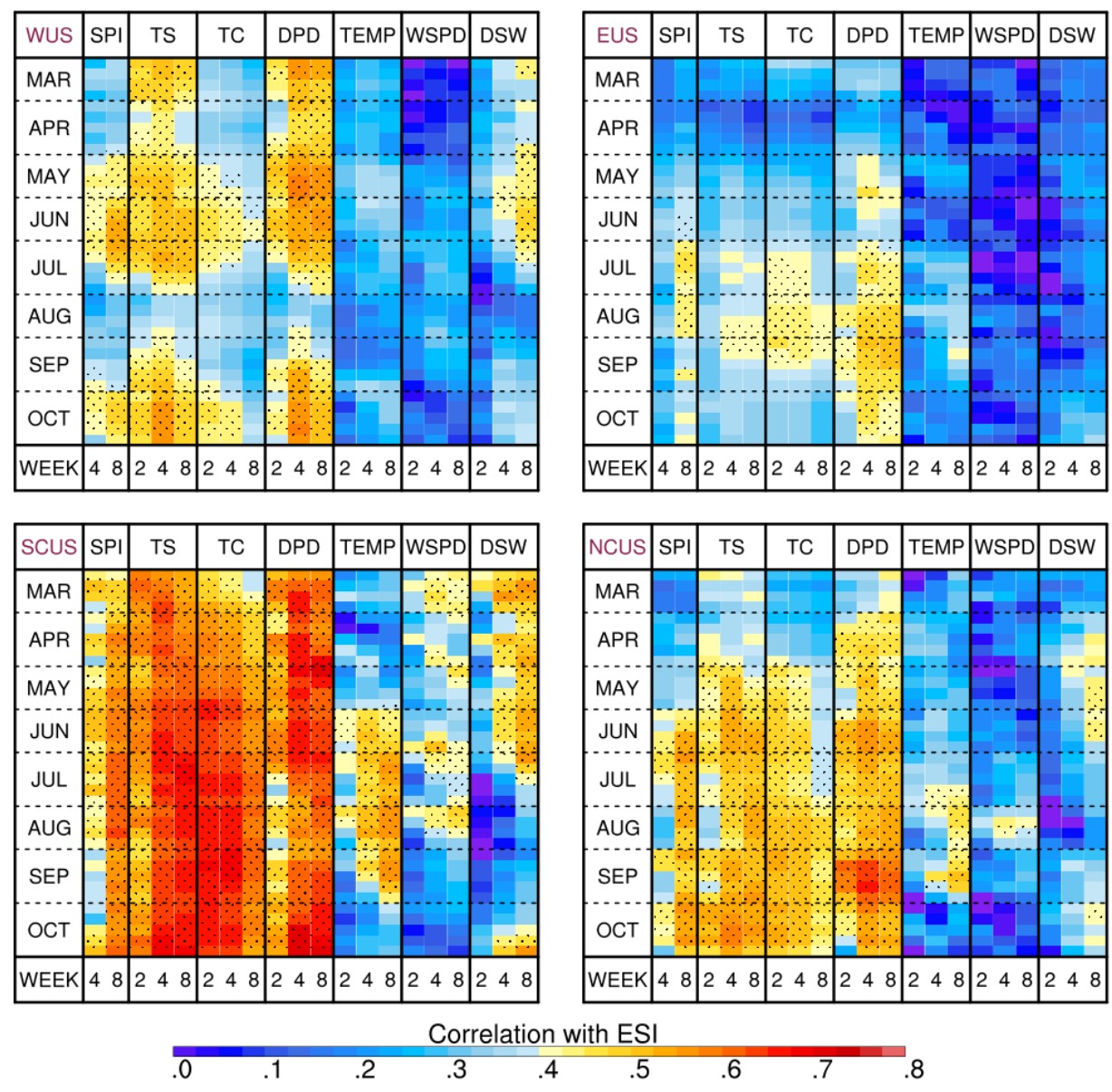

**Figure 5.** Same as Fig. 4, except for showing correlations computed with respect to 4-wk ESI anomalies.

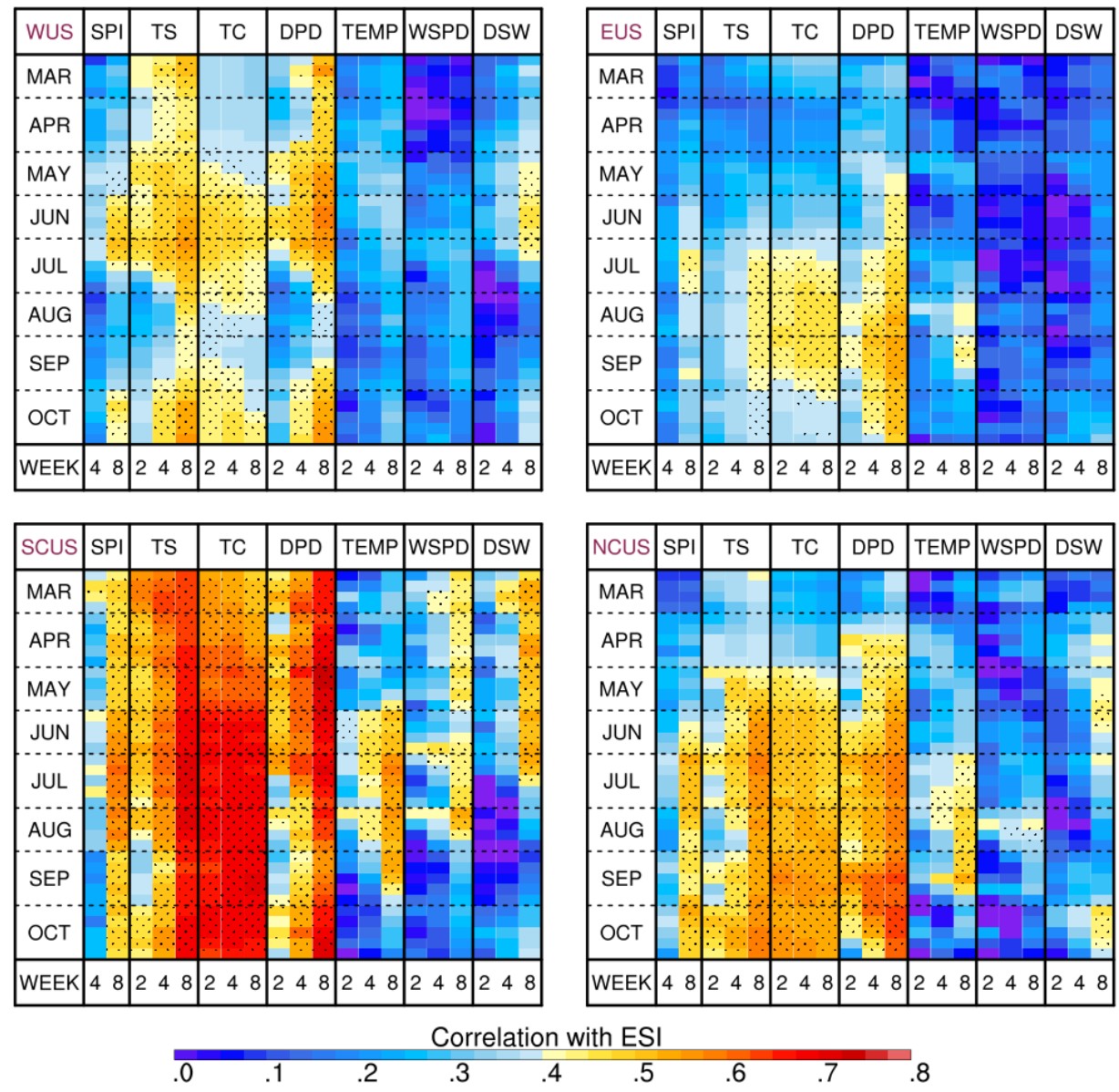

**Figure 6.** Same as Fig. 4, except for showing correlations computed with respect to 8-wk ESI anomalies.