# Peer review of "Exploring seasonal and regional relationships between the Evaporative Stress Index and surface weather and soil moisture anomalies across the United States"

_Hydrology and Earth System Sciences, 2018_

## Referee Comment (RC1) · Anonymous Referee #1 · 2 May 2018

This manuscript uses a correlation analysis to look at relationships between the Evaporative Stress Index (ESI) and a number of variables the effect evapotranspiration (ET) and resultant ESI values. Different composting time periods were used (2-, 4-, and 8-weeks). The main results find that soil moisture and dew point depression more strongly related to ESI than temperature and precipitation. The authors state this could be potentially useful in forecasting changes in ESI and vegetative stress.

This is a nice study overall with a straightforward and easy to interpret analysis. These results will be of interest to the drought and agricultural community for detecting early

signs of flash droughts. The main issue with the paper is that there are several key references missing. Many times throughout the paper it is stated that changes in evaporative demand (E0) drive changes in ET and ESI yet E0 is never used in the analysis. The Evaporative Demand Drought Index (EDDI) is also never used or cited, yet this analysis essentially repeats the analysis from the EDDI paper. References about dominant drivers of E0 are also needed here. With a more thorough literature review and acknowledgement of prior analysis that find similar conclusions, along with some other modifications this paper could be acceptable for publication.

Comments:

1. My biggest concern is failure to acknowledge EDDI in this study. In particular, EDDI part II (McEvoy et al. 2016). This analysis essentially repeats that analysis and finds many of the same conclusions without reference to the paper. Figures 1 and 2 are basically the same as Figures 1 and 4 in McEvoy et al. (2016). Figure 4 in McEvoy et al. (2016) actually directly compares ESI with E0 (EDDI). Figures 4-6 are also very similar to Figure 3 in McEvoy et al. (2016) regarding the regional analysis and many of the same conclusions are found regarding poor correlations in the northeast. The main reason this is all so concerning is that the authors state many times that E0 is driving ET and ESI yet fail to recognize recent literature that has already show very similar results. This paper should be framed as building upon other work that has done similar exercises. With that said, the results in this manuscript do add nice additional insight around the individual monthly correlations rather than the seasonal means used in McEvoy et al. (2016), and around some of the individual drivers of E0.

McEvoy, D. J., Huntington, J. L., Hobbins, M. T., Wood, A., Morton, C., Anderson, M., & Hain, C. (2016). The evaporative demand drought index. Part II: CONUS-wide assessment against common drought indicators. Journal of Hydrometeorology, 17(6), 1763-1779.

2. The other key reference missing is regarding sensitivity of E0 to the individual drivers

over CONUS. Hobbins (2016) describes this nicely. The dominant drivers of E0 are different depending on region and season. And, a key point of Hobbins (2016) finds that temperature is not the dominant driver for many regions. This is likely having a big influence on the how ESI (with E0 as one part of the ratio) is correlating to the different drivers. There should be some discussion around this reference.

Hobbins, M. T. (2016). The variability of ASCE standardized reference evapotranspiration: A rigorous, CONUS-wide decomposition and attribution. Transactions of the ASABE, 59(2), 561-576.

3. How was CFSR chosen as the data set to be used in the analysis? Were any other data sets tested? I would think, at least for temperature and humidity, that other finer scale station-based data sets would be better. My first thoughts for other choices would be PRISM or gridMET (http://www.climatologylab.org/gridmet.html). For wind and solar a regridded CFSR would probably be just fine. Can you provide any justification (in the manuscript or response to review) for choice of data? 4. The standardization of different variables appears to have all been done using different periods of record. SPI: 1948-2015, ESI: 2001-2015, NLDAS: 1979-2015, CFSR: 1979-2015. I would think using the 2001-2015 period for all variables (standardization, not just comparisons) would provide the most robust comparison. Levels of wet and dryness will not be comparable using different time periods. Can you justify the reasoning for using different periods of record? 5. How were the regions in Figure 3 defined? Seems very arbitrary. Something like the NCEI climate regions would probably be better. Please provide more detail on how/why these specific regions were used. 6. The method for the correlation analysis is a bit unclear. Please add a summary of 1-2 sentences about the sample size for the time series at each grid point. This is the part that is not clear to me. For a given grid point and month, would there be 15 years x 4 weekly values (n=60)?
* * *

---

## Referee Comment (RC2) · Anonymous Referee #2 · 18 Jul 2018

The manuscript presents a straighforward, correlation-based analysis of the linear relationship between the Evaporative Stress Index and a variety of land and atmosphere variables. The methods are simple and robust, and the results and conclusions of the paper are relevant and impactful, particularly for the drought monitoring/forecasting community. I have a few questions and issues with the manuscript in its current form, but I recommend accepting the paper for publication once these issues are addressed.

Specific Comments: 1) The main issue I have with the study is the use of CFSR without ample justification. Because the atmospheric variables are a key part of

the study, more details are needed for CFSR. Which of the variables used are observations that are assimilated into CFSR and which are modeled. Did you perform any kind of data verification or comparison with actual observations to ensure data fidelity? NCAR's CFSR data page (https://climatedataguide.ucar.edu/climate-data/climate-forecast-system-reanalysis-cfsr) actually lists "performance not well-known" as a key limitation of CFSR. This is in contrast to well-validated reanalysis datasets like ERA-Interim and NARR, or observation-based products like PRISM and the GHCN gridded products. Therefore, I recommend the authors either undertake a limited CFSR data validation with observations of TEMP, WSPD, etc., or repeat the correlation analyses using a dataset independent of CFSR, to ensure the results presented here are robust.

2) I'm left a little confused by the correlation method description. How many individual (e.g.,) ESI-SPI or ESI-WSPD points were included in the correlation for each month/year? For example, were there 4 pairs of ESI-SPI data points for May, 2008 or just 1 pair of points for May, 2008?

3) The correlation maps/figures have no indication of statistical significance. Could you perhaps show any area in which the correlation was not significant at (e.g.,) 90% or 95% confidence level as white instead of red or blue? Or maybe contour around areas in which the correlations are significant at some predetermined confidence level?
* * *

---

## Author Response (AR1)

This manuscript uses a correlation analysis to look at relationships between the Evaporative Stress Index (ESI) and a number of variables the effect evapotranspiration (ET) and resultant ESI values. Different composting time periods were used (2-, 4-, and 8-weeks). The main results find that soil moisture and dew point depression more strongly related to ESI than temperature and precipitation. The authors state this could be potentially useful in forecasting changes in ESI and vegetative stress.

This is a nice study overall with a straightforward and easy to interpret analysis. These results will be of interest to the drought and agricultural community for detecting early signs of flash droughts. The main issue with the paper is that there are several key references missing. Many times throughout the paper it is stated that changes in evaporative demand (E0) drive changes in ET and ESI yet E0 is never used in the analysis. The Evaporative Demand Drought Index (EDDI) is also never used or cited, yet this analysis essentially repeats the analysis from the EDDI paper. References about dominant drivers of E0 are also needed here. With a more thorough literature review and acknowledgement of prior analysis that find similar conclusions, along with some other modifications this paper could be acceptable for publication.

Comments:

1. My biggest concern is failure to acknowledge EDDI in this study. In particular, EDDI part II (McEvoy et al. 2016). This analysis essentially repeats that analysis and finds many of the same conclusions without reference to the paper. Figures 1 and 2 are basically the same as Figures 1 and 4 in McEvoy et al. (2016). Figure 4 in McEvoy et al. (2016) actually directly compares ESI with E0 (EDDI). Figures 4-6 are also very similar to Figure 3 in McEvoy et al. (2016) regarding the regional analysis and many of the same conclusions are found regarding poor correlations in the northeast. The main reason this is all so concerning is that the authors state many times that E0 is driving ET and ESI yet fail to recognize recent literature that has already show very similar results. This paper should be framed as building upon other work that has done similar exercises. With that said, the results in this manuscript do add nice additional insight around the individual monthly correlations rather than the seasonal means used in McEvoy et al. (2016), and around some of the individual drivers of E0.

McEvoy, D. J., Huntington, J. L., Hobbins, M. T., Wood, A., Morton, C., Anderson, M., & Hain, C. (2016). The evaporative demand drought index. Part II: CONUS-wide assessment against common drought indicators. Journal of Hydrometeorology, 17(6), 1763-1779.

*We apologize for not including this important reference in the original manuscript. We now mention this paper in the last paragraph of the introduction and have added a new paragraph to the conclusions section that compares results from both studies.*

2. The other key reference missing is regarding sensitivity of E0 to the individual drivers over CONUS. Hobbins (2016) describes this nicely. The dominant drivers of E0 are different depending on region and season. And, a key point of Hobbins (2016) finds that temperature is not the dominant driver for many regions. This is likely having a big influence on the how ESI (with E0 as one part of the ratio) is correlating to the different drivers. There should be some discussion around this reference.

Hobbins, M. T. (2016). The variability of ASCE standardized reference evapotranspiration: A rigorous, CONUS-wide decomposition and attribution. Transactions of the ASABE, 59(2), 561-576.

*We apologize for not including this important reference in the original manuscript. It is now properly referenced in the last paragraph of the introduction as well as in the conclusions section in the revised manuscript. It is also mentioned in Section 3.2 when providing justification for the analysis regions used during this study.*

3. How was CFSR chosen as the data set to be used in the analysis? Were any other data sets tested? I would think, at least for temperature and humidity, that other finer scale station-based data sets would be better. My first thoughts for other choices would be PRISM or gridMET (http://www.climatologylab.org/gridmet.html). For wind and solar a regridded CFSR would probably be just fine. Can you provide any justification (in the manuscript or response to review) for choice of data?

*We chose to use the CFSR atmospheric dataset for this project because it is the dataset used by the ALEXI model. No other datasets were tested; however, it is anticipated that the correlations will have similar patterns if other datasets were used.*

4. The standardization of different variables appears to have all been done using different periods of record. SPI: 1948-2015, ESI: 2001-2015, NLDAS: 1979-2015, CFSR: 1979-2015. I would think using the 2001-2015 period for all variables (standardization, not just comparisons) would provide the most robust comparison. Levels of wet and dryness will not be comparable using different time periods. Can you justify the reasoning for using different periods of record?

*We agree that the use of different periods of records for the datasets introduces some uncertainty to the absolute magnitude of the correlations; however, we feel that this approach is justified because it is consistent with prior studies by the authors. One of the main motivations is that it takes at least a 30-year record to compute realistic SPI anomalies. Because the ESI dataset only covers a 15-year period, this means that we would be forced to use different periods of record for these datasets anyway, which makes it more attractive to simply use the full period of records for each dataset.*

5. How were the regions in Figure 3 defined? Seems very arbitrary. Something like the NCEI climate regions would probably be better. Please provide more detail on how/why these specific regions were used.

*Several sentences were added to the first paragraph in the regional correlation analysis section that provide justification for these regions:*

*These regions were chosen based on geography, climate, and inspection of the spatial patterns in the correlations found in Figs. 1 and 2. In particular, the central U.S. regions encompass the meridional gradient between more arid climates to the west and more humid climates to the east (Seager et al. 2018). This area was further separated into south-central and north-central regions to highlight the stronger correlations over the south-central U.S. and to account for large regional differences in the variance of the atmospheric drivers of reference ET noted by Hobbins (2016). Likewise, remaining areas were simply grouped into western and eastern regions that generally encompass more arid and more humid climates, respectively.*

6. The method for the correlation analysis is a bit unclear. Please add a summary of 1-2 sentences about the sample size for the time series at each grid point. This is the part that is not clear to me. For a given grid point and month, would there be 15 years x 4 weekly values (n=60)?

*A sentence was added to the first paragraph of the monthly correlation analysis section that states: "This means that the sample size (n) for each grid point is equal to 60 or 75 depending upon whether a given month contains the end dates for four or five of these 4-week periods."*
* * *
**Anonymous Referee #2**

The manuscript presents a straightforward, correlation-based analysis of the linear relationship between the Evaporative Stress Index and a variety of land and atmosphere variables. The methods are simple and robust, and the results and conclusions of the paper are relevant and impactful, particularly for the drought monitoring/forecasting community. I have a few questions and issues with the manuscript in its current form, but I recommend accepting the paper for publication once these issues are addressed.

Specific Comments:

1) The main issue I have with the study is the use of CFSR without ample justification. Because the atmospheric variables are a key part of the study, more details are needed for CFSR. Which of the variables used are observations that are assimilated into CFSR and which are modeled. Did you perform any kind of data verification or comparison with actual observations to ensure data fidelity? NCAR's CFSR data page (https://climatedataguide.ucar.edu/climatedata/climate-forecast-system-reanalysis-cfsr) actually lists "performance not well known as a key limitation of CFSR. This is in contrast to well-validated reanalysis datasets like ERA-Interim and NARR, or

observation-based products like PRISM and the GHCN gridded products. Therefore, I recommend the authors either undertake a limited CFSR data validation with observations of TEMP, WSPD, etc., or repeat the correlation analyses using a dataset independent of CFSR, to ensure the results presented here are robust.

*We chose to use the CFSR atmospheric dataset for this project because it is the dataset used by the ALEXI model. We agree that the absolute magnitudes of the correlations could depend upon which reanalysis or observation datasets are used; however, we anticipate that the regional and seasonal correlation patterns along with the relative importance of each variable will remain similar regardless of which datasets are used.*

*We agree with your sentiment that more verification studies are necessary; however, that work is beyond the scope of the current project. Several sentences were added to the section describing the CFSR dataset that discuss these variables, cite verification studies, and present an advantage to using this dataset:*

*"It is important to note that though all of these variables are derived from model output, they are constrained through the assimilation of satellite and conventional observations within the CFSR data assimilation system. Regional verification studies, such as those performed by Lindsay et al. (2014) and Sharp et al. (2015), have shown that the accuracy of the CFSR near-surface variables are comparable to those from other reanalysis datasets. The use of reanalysis data introduces some uncertainty to the evaluation performed during this study but it has the advantage of providing uniform spatial resolution across the entire region."*

2) I'm left a little confused by the correlation method description. How many individual (e.g.,) ESI-SPI or ESI-WSPD points were included in the correlation for each month/year? For example, were there 4 pairs of ESI-SPI data points for May, 2008 or just 1 pair of points for May, 2008?

*You are correct to note that there would be 4 or 5 pairs of data points for each month (or alternatively 60 or 75 per month for the full period of record). To make this more explicit, a sentence was added to the first paragraph of the monthly correlation analysis section that states: "This means that the sample size (n) for each grid point is equal to 60 or 75 depending upon whether a given month contains the end dates for four or five of these 4-week periods."*

3) The correlation maps/figures have no indication of statistical significance. Could you perhaps show any area in which the correlation was not significant at (e.g.,) 90% or 95% confidence level as white instead of red or blue? Or maybe contour around areas in which the correlations are significant at some predetermined confidence level?

*As suggested, we computed the statistical significance at each grid point. This was done using the "rtest" routine in the NCAR Command Language (NCL) package. Figures 1 and 2 were subsequently modified so that only those grid points significant at the p=0.1 level were plotted. For Figs. 4-6, stippling was added to denote 
[revised manuscript text omitted]

---

## Referee Report (RR1)

Overall, the authors have done fair job addressing my comments. However, there are a couple issues that I am not completely satisfied with.

1. The issues of how/why CFSR was used has now been brought up by both reviewers and the editor. The editor has actually suggested a quick comparison between CFSR and other data sets and reviewer #2 has also suggested this. I agree that the overall spatial patterns of correlations *might* not change, but we don't know this for sure. I would also encourage the authors to provide a basic analysis to justify data choice. Simply stating that CFSR is used with ALEXI does not seem like enough.

2. I have to disagree with the authors regarding what period of record to use for standardization (see reviewer 1 comment #4). The authors state in the reply that it takes at least 30-years (that number is debatable to this day) for a stable SPI. So wouldn't at least 30 years be needed for a stable ESI? My main argument is that trends in the individual drivers will have a big impact on normalization when using different periods of record. This will be particularly important for things like temperature and dew point (and any other variables that are related to temperature). A "low" normalized temperature value with a record of 2001-2015 may not be low at all when using the 1979-2015 record. We all know the climate has been changing dramatically over the past several decades, so this is a big deal to make the study robust. For me, these periods need to be consistent for the paper to be published.

I will recommend the paper for publication once the normalization periods are consistent. The CFSR issue probably should be addressed but I realize that is a lot of extra work outside the scope of study. The authors also provide two new references pointing at CFSR validation studies.

---

## Author Response (AR2)

Overall, the authors have done fair job addressing my comments. However, there are a couple issues that I am not completely satisfied with.

1. The issues of how/why CFSR was used has now been brought up by both reviewers and the editor. The editor has actually suggested a quick comparison between CFSR and other data sets and reviewer #2 has also suggested this. I agree that the overall spatial patterns of correlations *might* not change, but we don't know this for sure. I would also encourage the authors to provide a basic analysis to justify data choice. Simply stating that CFSR is used with ALEXI does not seem like enough.

*The CFSR is a high-quality reanalysis dataset that has been used by many studies and represents an important improvement over previous generations of reanalysis datasets (such as NARR). In our previous revision, we added two references that showed that CFSR surface reanalyses are of comparable quality to other reanalysis datasets. We have added three more references to this paragraph that showed similar conclusions. The revised manuscript now states:*

*"Regional verification studies, such as those performed by Bao and Zhang (2013), Lindsay et al. (2014), Sharp et al. (2015) and Essou et al. (2016), have shown that the accuracy of the CFSR near-surface variables are comparable to those from other reanalysis datasets and represent an important improvement over previous generations of reanalysis datasets. Fuka et al. (2013) have shown that when CFSR data was used to force a watershed model, that it produced stream discharge simulations that were as good or better than models forced using weather station observations. The use of reanalysis data introduces some uncertainty to the evaluation performed during this study but it has the advantage of providing uniform spatial resolution across the entire region."*

*We agree with you that there could be some sensitivity in the absolute magnitude of the correlations to the choice of reanalysis datasets; however, we reiterate that this is not expected to have a material impact on the regional and seasonal correlation patterns or on the relative importance of each variable. Because other studies have shown that the CFSR surface reanalyses have comparable accuracy to other reanalysis datasets, we hope that the reviewers will be satisfied with these revisions.*

2. I have to disagree with the authors regarding what period of record to use for standardization (see reviewer 1 comment #4). The authors state in the reply that it takes at least 30-years (that number is debatable to this day) for a stable SPI. So wouldn't at least 30 years be needed for a stable ESI? My main argument is that trends in the individual drivers will have a big impact on normalization when using different periods of record. This will be particularly important for things like temperature and dew point (and any other variables that are related to temperature). A "low" normalized temperature value with a record of 2001-2015 may not be low at all when using the 1979-2015 record. We all know the climate has been changing dramatically over the past several decades, so this is a big deal to make the study robust. For me, these periods need to be consistent for the paper to be published.

*To address the reviewer's comment, we computed the correlations between the 4-week ESI and 4-week SPI when the SPI anomalies were computed using data either from the full period of record for the SPI (1948-2016) or from the much shorter ESI period of record (2001-2016). In both cases, the correlations were computed using data from 2001-2016. The correlations between each dataset, along with their differences, are shown in the figure below. Overall, it is apparent that the correlation pattern is robust and that there are no discernable differences in the correlations across the U.S. Because this example represents the most extreme case in terms of differences in periods of records between the various datasets, and the differences between these analyses were very small, we expect the differences for other variable combinations to be of similar or less magnitude. Because of this, we do not think that it is necessary to redo the entire analysis so that the same periods of records are used for all datasets.*

[Figure]

[Figure]

[Figure]

I will recommend the paper for publication once the normalization periods are consistent. The CFSR issue probably should be addressed but I realize that is a lot of extra work outside the scope of study. The authors also provide two new references pointing at CFSR validation studies.

*We have performed additional analysis and have concluded that it is not necessary to redo the entire analysis (see above comment). We hope that the reviewers agree with this conclusion.*

---

## Author Response (AR3)

The authors have provided an example showing SPI correlations using only the 2001-2016 period and it shows minimal differences in spatial patterns compared to the SPI computed using 1948-2016. However, my main concern was with the temperature related variables, which is not shown.

The manuscript should be accepted for publication after the authors add one or two sentences stating some uncertainty around the correlation due to different periods of record used, particularly with temperature related variables (since temperature was not shown in the authors response I'm not yet completed convinced.

*Two sentences were added to section 2.4 to discuss this uncertainty.*